# Phenyl Derivatives Modulate the Luminescent Properties and Stability of CzBTM-Type Radicals

**DOI:** 10.3390/molecules29122900

**Published:** 2024-06-18

**Authors:** Quanquan Gou, Jiahao Guan, Lintao Zhang, Xin Ai

**Affiliations:** 1School of Materials Science and Engineering, Hainan University, No 58, Renmin Avenue, Haikou 570228, China; 2Collaborative Innovation Center of Marine Science and Technology, Hainan University, No 58, Renmin Avenue, Haikou 570228, China; 3Collaborative Innovation Center of Information Technology, Hainan University, No 58, Renmin Avenue, Haikou 570228, China

**Keywords:** organic luminescent radical, CzBTM, blueshift, stability

## Abstract

The distinctive electron structures of luminescent radicals offer considerable potential for a diverse array of applications. Up to now, the luminescent properties of radicals have been modulated through the introduction of electron-donating substituents, predominantly derivatives of carbazole and polyaromatic amines with more and more complicated structures and redshifted luminescent spectra. Herein, four kinds of (*N*-carbazolyl)bis(2,4,6-tirchlorophenyl)-methyl (CzBTM) radicals, **Ph_2_CzBTM**, **Mes_2_CzBTM**, **Ph_2_PyIDBTM**, and **Mes_2_PyIDBTM**, were synthesized and characterized by introducing simple phenyl and 2,4,6-trimethylphenyl groups to CzBTM and PyIDBTM. These radicals exhibit rare blueshifted emission spectra compared to their parent radicals. Furthermore, modifications to CzBTM significantly enhanced the photoluminescence quantum yields (PLQYs), with a highest PLQY of 21% for **Mes_2_CzBTM** among CzBTM-type radicals. Additionally, the molecular structures, photophysical properties of molecular orbitals, and stability of the four radicals were systematically investigated. This study provides a novel strategy for tuning the luminescent color of radicals to shorter wavelengths and improving thermostability.

## 1. Introduction

The unique photophysical properties of organic radicals, arising from their specific unpaired electron structure [1,2,3,4], have facilitated their application in various fields, including organic field-effect transistors (OFETs) [5,6], organic light-emitting diodes (OLEDs) [7,8,9,10,11], magneto-luminescence [12,13,14,15], spin manipulation [16,17,18], and sensors [19,20,21,22,23]. Thus, more and more researchers have been attracted to developing high-performance luminescent radicals. Generally, in most studies, the luminescent efficiency of radicals was improved by the introduction of electron-donating substituents. Substituent groups which were incorporated into radicals range from simple donors like indole [14], carbazole [24], and triphenylamine [25,26] to much more complicated substituted carbazole derivatives [27,28] and even dendritic derivatives [29]. It has been demonstrated that the above strategy was indeed an effective way to improve the luminescent property and stability of radicals but also would result in redshifts of emission spectra.

To date, for classic luminescent radical systems, like the tris(2,4,6-trichlorophenyl)methyl (TTM), (3,5-Dichloro-4-pyridyl)bis(2,4,6-trichlorophenyl)methyl (PyBTM), and (*N*-carbazolyl)bis(2,4,6-trichlorophenyl)-methyl (CzBTM) radicals, blueshifts in the emission spectrum through chemical modification is challenging. Among the radical systems mentioned above, the only efficient reported strategy to shift the emission spectrum toward shorter wavelengths was the replacement of chloride atoms by more electronegative fluorine atoms [30]. However, this strategy fails when applied to the carbazole group in CzBTM. Not only the electronegativity but also the torsion angle of carbazole affects the change in the emission spectrum [31,32]. Another successful approach to obtain blueshifted emission spectra is the introduction of a nitrogen atom into the carbazolyl moiety of CzBTM, which leads to a weakening of the electron-donating property of carbazolyl [33]. Herein, a different strategy was carried out to change the performance of CzBTM-type radicals. Given that CzBTM is itself a donor–acceptor molecule, unlike the changes in the carbazole moiety from previous research, the modification on the radical side could also affect the luminescent properties. In this paper, weak electron-donating groups, phenyl and 2,4,6-trimethylphenyl, were introduced into the radical side of CzBTM and its analogue PyIDBTM (Figure 1). The molecular structures, photophysical properties, and stabilities of these derivatives were systematically investigated. The introduction of additional proper donors at the radical sides in the four derivatives could weaken the electron-accepting ability to the radical, resulting in blueshifts of the emission spectra. In addition, this molecular design strategy could also improve the photoluminescence quantum yield (PLQY) and thermostability of CzBTM-type radicals.

## 2. Results and Discussion

### 2.1. Synthesis and Structure Characterization

CzBTM and PyIDBTM were synthesized from commercially available reagents and the methods reported in the literature were followed [31,33]. Four target radicals, **Ph_2_CzBTM**, **Mes_2_CzBTM**, **Ph_2_PyIDBTM,** and **Mes_2_PyIDBTM,** were synthesized utilizing the Suzuki–Miyaura coupling reaction as the final step in yields of 30%, 23%, 21%, and 15%, respectively. The synthetic details are outlined in the Supporting Information (Appendix A). The molecular structures of the four radicals were confirmed by MALDI-TOF mass spectra and Fourier transform infrared spectra (FT-IR) at first (Appendix A). The existence of the unpaired electron, so as the radical nature, was confirmed by electron paramagnetic resonance (EPR) spectra at room temperature, with *g* values of 2.0037, 2.0038, 2.0021, and 2.0021 (Figure 2a–d).

To further explore the molecular structures of these four radical molecules, attempts were made to gain single crystals. However, up to now, only ideal single crystals of the **Ph_2_CzBTM** and **Mes_2_CzBTM** radicals were obtained using the slow diffusion method with dichloromethane/ethanol mixed solution at room temperature. As for the **Ph_2_PyIDBTM** and **Mes_2_PyIDBTM** radicals, polycrystal samples were always obtained. Thus, only the structures of **Ph_2_CzBTM** and **Mes_2_CzBTM** were further determined through single-crystal X-ray diffraction measurements (Figure 2e,f). The solved structures reveal that the central methyl carbon atoms, C13 in both molecules, are sp^2^-hybridized and are coplanar with N1, C14, and C26 (or C29), which also confirmed the existence of an unpaired electron. The torsion angles of the dichlorophenyl rings and carbazole group to the radical center plane are approximately 45.7° and 47.7° (N1-C13-C14-C15 and C1-N1-C13-C14) in **Ph_2_CzBTM**, as well as 48.6° and 35.6° (N1-C13-C14-C15 and C1-N1-C13-C14) in **Mes_2_CzBTM**. The dihedral angles between the benzene rings and dichlorophenyl groups are 32.9° and 48.7° (C16-C17-C20-C21 and C28-C29-C32-C33) in **Ph_2_CzBTM** and significantly increase to 93.5° and 70° (C16-C17-C20-C21 and C31-C32-C35-C36) when more sterically hindered 2,4,6-trimethylphenyl groups were introduced in **Mes_2_CzBTM**. Multiple intermolecular interactions between adjacent molecules could be found in the crystals of two radicals (Appendix A).

### 2.2. Photophysical Properties

The ultraviolet–visible (UV-Vis) absorption spectra and photoluminescence (PL) spectra of **Ph_2_CzBTM**, **Mes_2_CzBTM**, **Ph_2_PyIDBTM,** and **Mes_2_PyIDBTM** were measured in cyclohexane solvent (Figure 3a,b). At the same time, the photophysical properties of CzBTM and PyIDBTM were also measured for comparison. Similar to CzBTM and PyIDBTM, **Ph_2_CzBTM**, **Mes_2_CzBTM**, **Ph_2_PyIDBTM,** and **Mes_2_PyIDBTM** also display three absorption bands, including a strong absorption band at 280 nm caused by the carbazole moiety or β-carboline, a medium absorption band around 400 nm attributed to the characteristic absorption of carbon-centered radicals, and a weak absorption band beyond 500 nm primarily attributed to the intramolecular charge-transfer (CT) state. However, in comparison to **Mes_2_CzBTM** and **Mes_2_PyIDBTM**, there is a significant redshift in the medium absorption band of **Ph_2_CzBTM** and **Ph_2_PyIDBTM**, which is attributed to the difference in their transition energy caused by the difference in their molecular structures. Compared to the substituents in **Mes_2_CzBTM** and **Mes_2_PyIDBTM**, the phenyls in **Ph_2_CzBTM** and **Ph_2_PyIDBTM** formed larger conjugated structures due to the smaller dihedral angles, resulting in changes in the characteristic absorption of the carbon-centered radicals, which is also confirmed by the subsequent theoretical calculations.

As for the PL spectra (Figure 3b), **Ph_2_CzBTM** and **Mes_2_CzBTM** exhibit a red emission that peaked at 683 nm and 672 nm, with blueshifts of about 30 nm and 41 nm, respectively, compared to CzBTM (713 nm). Similar blueshifts could also be observed in the PyIDBTM series. The emission peak of **Ph_2_PyIDBTM** is at 653 nm, while that of **Mes_2_PyIDBTM** is at 630 nm, showing blueshifts of 11 nm and 34 nm, respectively, compared to PyIDBTM (664 nm). These results demonstrated that the molecular design strategy in this work is effective to realize blueshifts of the emission spectra of CzBTM-type radicals. And 630 nm for the emission peak is the shortest wavelength for the CzBTM derivatives reported. All the radicals show nearly unchanged absorption peaks in the solvents of different polarities, but the PL spectra undergo a significant redshift with increasing solvent polarity, which demonstrated the CT excited-state nature (Appendix A).

The absolute PLQY (*Φ_f_*) of **Ph_2_CzBTM** and **Mes_2_CzBTM** in cyclohexane solution is 9.9% and 21.0%, respectively, which are obviously higher than that of CzBTM (*Φ_f_* = 2.0%). Meanwhile, the *Φ_f_* of **Ph_2_PyIDBTM** and **Mes_2_PyIDBTM** is 6.4% and 9.8%, respectively, without improvement compared to the parent molecule but also better than CzBTM. The transient photoluminescence decay spectra of the four radicals in cyclohexane, as shown in Appendix A, exhibit single exponential decays with lifetimes (*τ*) of 9.3 ns and 15.2 ns, respectively. Additionally, the lifetimes of **Ph_2_PyIDBTM** and **Mes_2_PyIDBTM** are 6.4 ns and 8.6 ns, respectively, which are slightly shorter than that of PyIDBTM (*τ* = 19.5 ns). Based on Equations (1) and (2), the radiative rate constant (*k_r_*) and non-radiative rate constant (*k_nr_*) for **Ph_2_CzBTM**, **Mes_2_CzBTM**, **Ph_2_PyIDBTM,** and **Mes_2_PyIDBTM** are calculated (Table 1).
(1)$$\Phi =\frac{k_{r}}{k_{r}+k_{nr}}$$
(2)$$\tau =\frac{1}{k_{r}+k_{nr}}$$

The results show that, compared to CzBTM, the *k_r_* values of the four derivatives have been enhanced, while the *k_nr_* significantly decreased. The shorter emission wavelengths of all the derivatives, which means a larger energy gap as confirmed by following the theoretical calculations, serves to diminish the Frank–Condon factors, implying a decline in the internal conversion [34] and consequently a decrease in the *k_nr_*. Additionally, due to the larger steric hindrance in **Ph_2_CzBTM** and **Mes_2_CzBTM**, effective inhibition of the molecular rotation and vibration occurs, also leading to a significantly smaller *k_nr_*. Thus, **Mes_2_CzBTM** exhibits the highest PLQY. However, the *k_nr_* values of **Ph_2_PyIDBTM** and **Mes_2_PyIDBTM** do not decrease further compared to PyIDBTM, which may be affected by the LE-dominated excited state of PyIDBTM in low-polarity solvent [35].

### 2.3. Theoretical Calculations

Density functional theoretical (DFT) calculations (B3LYP/6-31G (d,p)) were performed to understand the frontier molecular orbitals (MOs) of CzBTM-type radicals. The results show that the spin density distributions (Appendix A) of **Mes_2_CzBTM** and **Mes_2_PyIDBTM** are similar to those of CzBTM and PyIDBTM, while the spin density distributions of **Ph_2_CzBTM** and **Ph_2_PyIDBTM** are partially delocalized to the benzene rings. Through the calculation MOs (Figure 4a), the single occupied molecular orbital (α-SOMO) and the single unoccupied molecular orbital (β-SUMO) are mainly distributed on the central carbon atom and partially extend to the dichlorophenyl rings and carbazole moiety for **Mes_2_CzBTM** (160α and 160β), while those of **Ph_2_CzBTM** (184α and 184β) are distributed throughout the molecular skeleton. The highest occupied β molecular orbital (β-HOMO) of **Mes_2_CzBTM** (183β) and **Ph_2_CzBTM** (159β) is mainly distributed on the dichlorophenyl rings and carbazole moiety, which is different from CzBTM. The lowest unoccupied β molecular orbital (β-LUMO) is mainly distributed on the dichlorophenyl and outer phenyl groups for **Ph_2_CzBTM** (161β). However, the β-LUMO of **Mes_2_CzBTM** (183β) is only located on the dichlorophenyl groups, which is caused by the large dihedral angle between the mesityl and dichlorophenyl groups. The distribution of the frontier electronic orbitals of PyIDBTM, **Ph_2_PyIDBTM,** and **Mes_2_PyIDBTM** are shown in Figure 4b, showing a similar characterization as discussed above. The energy levels of the frontier MOs are summarized in Appendix A. As can be seen in Figure 4, the energy levels of the α-SOMO and β-SUMO of **Ph_2_CzBTM** (−4.67 eV and −2.74 eV) and **Mes_2_CzBTM** (−4.71 eV and −2.67 eV) are higher than that of CzBTM (−4.99 eV and −3.01 eV). Similarly, the energy levels of **Ph_2_PyIDBTM** and **Mes_2_PyIDBTM** are also higher than that of PyIDBTM. From the above results, it can be understood that when the phenyl derivatives are introduced, the energy levels of the HOMO, α-SOMO, and β-SUMO obviously increase. However, for the donor itself, the carbazole or β-carboline do not change. Thus, the phenyl derivatives mainly caused the increase in the MO energy levels of the diphenyl methyl radical side, which is also equivalent to the weakening of the electron-accepting ability to the radical. As a result, the PL spectra show obviously blue shifts.

Additionally, time-dependent DFT (TD-DFT) calculations (B3LYP/6-31G (d,p)) were carried out to further investigate the excited states. The results show that the doublet excited states (D_1_) (Appendix A) of **Ph_2_CzBTM**, **Mes_2_CzBTM**, **Ph_2_PyIDBTM,** and **Mes_2_PyIDBTM** originate from 159β→160β, 183β→184β, 159β→160β, and 183β→184β transitions, respectively. Furthermore, the transition energies of D_1_ were calculated, with the CzBTM radicals and derivatives as 2.19 eV, 2.21 eV, and 2.27 eV (CzBTM < **Ph_2_CzBTM** < **Mes_2_CzBTM**) and the PyIDBTM radicals and derivatives as 2.27 eV, 2.28 eV, and 2.36 eV (PyIDBTM < **Ph_2_PyIDBTM** < **Mes_2_PyIDBTM**). This further confirms the blueshift observed in the PL spectra.

### 2.4. Electrochemical Properties

The electrochemical properties of **Ph_2_CzBTM**, **Mes_2_CzBTM**, **Ph_2_PyIDBTM,** and **Mes_2_PyIDBTM** were measured via the cyclic voltammetry (CV) method in CH_2_Cl_2_ with a 0.1 M TBAPF_6_ solution serving as the supporting electrolyte. Similar to CzBTM and PyIDBTM, **Ph_2_CzBTM**, **Mes_2_CzBTM**, **Ph_2_PyIDBTM,** and **Mes_2_PyIDBTM** exhibit reversible oxidation and reduction processes (Appendix A). Utilizing Fc^+^/Fc as reference, the energy levels of the α-SOMO of **Ph_2_CzBTM**, **Mes_2_CzBTM**, **Ph_2_PyIDBTM,** and **Mes_2_PyIDBTM** are calculated to be −4.49 eV (*E*_ox_ = −0.31 V), −4.50 eV (*E*_ox_ = −0.30 V), −4.93 eV (*E*_ox_ = 0.13 V), and −4.95 eV (*E*_ox_ = 0.15 V) from the oxidation potentials, which are higher than the α-SOMO levels of CzBTM (−4.74 eV, *E*_ox_ = −0.06 V) and PyIDBTM (−4.83 eV, *E*_ox_ = 0.03 V). The β-SUMO energy levels of **Ph_2_CzBTM**, **Mes_2_CzBTM**, **Ph_2_PyIDBTM,** and **Mes_2_PyIDBTM** are calculated to be −3.59 eV (*E*_red_ = −1.21 V), −3.49 eV (*E*_red_ = −1.31 V), − 3.96 eV (*E*_red_ = −0.84 V), and −3.83 eV (*E*_red_ = −0.97 V) from the reduction potentials, which are also higher than CzBTM (−3.88 eV, *E*_red_ = −0.92 V) and PyIDBTM (−3.77 eV, *E*_ox_ = −1.03 V). These data also suggest that incorporating phenyl derivatives would increase the frontier energy levels of radicals, which is consistent with the above theoretical calculations. A total of 20 cycles of CV scans were also performed on **Ph_2_CzBTM**, **Mes_2_CzBTM**, **Ph_2_PyIDBTM**, and **Mes_2_PyIDBTM**, and these redox peaks remained unchanged (Appendix A).

The electrochemical properties of **Ph_2_CzBTM**, **Mes_2_CzBTM**, **Ph_2_PyIDBTM,** and **Mes_2_PyIDBTM** were measured via the cyclic voltammetry (CV) method in CH_2_Cl_2_ with a 0.1 M TBAPF_6_ solution serving as the supporting electrolyte. Similar to CzBTM and PyIDBTM, **Ph_2_CzBTM**, **Mes_2_CzBTM**, **Ph_2_PyIDBTM,** and **Mes_2_PyIDBTM** exhibit reversible oxidation and reduction processes (Appendix A). Utilizing Fc^+^/Fc as reference, the energy levels of the α-SOMO of **Ph_2_CzBTM**, **Mes_2_CzBTM**, **Ph_2_PyIDBTM,** and **Mes_2_PyIDBTM** are calculated to be −4.49 eV (*E*_ox_ = −0.31 V), −4.50 eV (*E*_ox_ = −0.30 V), −4.93 eV (*E*_ox_ = 0.13 V), and −4.95 eV (*E*_ox_ = 0.15 V) from the oxidation potentials, which are higher than the α-SOMO levels of CzBTM (−4.74 eV, *E*_ox_ = −0.06 V) and PyIDBTM (−4.83 eV, *E*_ox_ = 0.03 V). The β-SUMO energy levels of **Ph_2_CzBTM**, **Mes_2_CzBTM**, **Ph_2_PyIDBTM,** and **Mes_2_PyIDBTM** are calculated to be −3.59 eV (*E*_red_ = −1.21 V), −3.49 eV (*E*_red_ = −1.31 V), − 3.96 eV (*E*_red_ = −0.84 V), and −3.83 eV (*E*_red_ = −0.97 V) from the reduction potentials, which are also higher than CzBTM (−3.88 eV, *E*_red_ = −0.92 V) and PyIDBTM (−3.77 eV, *E*_ox_ = −1.03 V). These data also suggest that incorporating phenyl derivatives would increase the frontier energy levels of radicals, which is consistent with the above theoretical calculations. A total of 20 cycles of CV scans were also performed on **Ph_2_CzBTM**, **Mes_2_CzBTM**, **Ph_2_PyIDBTM**, and **Mes_2_PyIDBTM**, and these redox peaks remained unchanged (Appendix A).

### 2.5. Stability

The thermostabilities of the **Ph_2_CzBTM**, **Mes_2_CzBTM**, **Ph_2_PyIDBTM,** and **Mes_2_PyIDBTM** were measured by thermogravimetric analysis (TGA) under ambient and nitrogen atmospheres (Figure 5). The decomposition temperatures (Td, corresponding to 5% weight loss) for the four radicals are 254 °C, 301 °C, 284 °C, and 303 °C, respectively, in an ambient atmosphere and 273 °C, 311 °C, 291 °C, and 301 °C, respectively, in a nitrogen atmosphere, which are obviously higher than those of CzBTM (230 °C) and PyIDBTM (241 °C). In particular, the introduction of 2,4,6-trimethylphenyl significantly increased the decomposition temperature of the radicals, effectively enhancing their thermostability. Two main factors contribute to this result: firstly, the substituents are introduced into the more reactive chlorine sites of the CzBTM and PyIDBTM radicals, reducing the reactivity; secondly, the introduction of phenyl derivatives increases the spatial steric hindrance of the molecules, reducing the molecular thermal vibrations and thereby improving the thermal stability of the radical molecules.

Photostability is another important property for luminescent radicals. The fluorescence intensity decays of the **Ph_2_CzBTM**, **Mes_2_CzBTM**, **Ph_2_PyIDBTM**, and **Mes_2_PyIDBTM** radicals were monitored during continuous irradiation from a xenon lamp, compared with the CzBTM radical and PyIDBTM. The results are summarized in Appendix A. The half-life (t_1/2_) values of the radicals were 5.4 × 10^4^ s (**Ph_2_CzBTM**), 2.3 × 10^5^ s (**Mes_2_CzBTM**), 1.4 × 10^6^ s (CzBTM), 4.1 × 10^4^ s (**Ph_2_PyIDBTM**), 2.3 × 10^3^ s (**Mes_2_PyIDBTM**), and 4.4 × 10^4^ s (PyIDBTM), respectively. The results indicate that the introduction of simple substituents does not enhance the photostability of the radicals because of the increase in the excited-state energy. However, compared to other luminescent radicals, these four radical molecules still exhibit relatively good photostability.

## 3. Materials and Methods

The detailed synthesis route of the target radicals can be found in Supporting Information 1.1. The raw materials and chemical reagents utilized in this investigation were procured from ERNEGI and Xilong Science Co., Ltd. (Shanghai, China), sans additional refinement. The infrared spectra were captured using a Bruker Tensor 27 spectrometer. The KBr and the radical were thoroughly mixed at a mass ratio of 100:1 and the mixture was ground well. Then, the mixture was pressed into a lamellar state for the IR measurements. The spectra of the NMR were obtained at room temperature on a Bruker AVANCE NEO 400. Deuterated chloroform was used as the solvent for both the ^13^C NMR and ^1^H NMR spectra. The mass spectra were primarily obtained through testing with two instruments: LCMS-IT-TOF and MALDI-TOF. DCTB was selected as the matrix for the MALDI-TOF testing. The EPR testing was conducted with a Bruker A320 spectrometer. The radical materials were tested at room temperature in both the solid state and solution (with dichloromethane as the solvent at a concentration of 0.1 M). The elemental analysis data were recorded on an Elementar Vario micro cube spectrometer. The single-crystal structure testing results were obtained using the Bruker D8 Quest instrument. The analysis of the single-crystal structures and their further analysis were conducted using the Olex2 software. The testing temperature for the single-crystal structures was maintained at 100 K. The UV–visible spectra were obtained using a Shimadzu UV-1900i UV-Vis spectrometer (Shimadzu Corporation, Kyoto, Japan). The photoluminescence spectra were acquired with a Shimadzu RF-6000 spectrometer (Shimadzu Corporation, Japan). The PL decays were captured employing an Edinburgh FLS1000 spectrometer (Edinburgh Instruments, Livingston, UK), and the absolute PLQYs were measured on the identical apparatus via the integrating sphere method. The DFT and TD-DFT calculations were executed using the Gaussian16 C.02 commercial software [36]. The TGA test results were obtained using the TA INSTRUMENTS Q600 instrument (New Castle, DE, USA), with the tests conducted on the radicals in both air and nitrogen atmospheres, at a heating rate of 10 °C/min. The cyclic voltammetry measurements were carried out with a CH Instruments CHI660E electrochemical analyzer (Austin, TX, USA). The results of the photostability were obtained through testing under continuous xenon lamp irradiation, using the Shimadzu RF-6000 spectrometer.

### 3.1. Synthesis of Ph_2_CzBTM and Mes_2_CzBTM

The specific synthetic route is detailed in Appendix A. **Ph_2_CzBTM** is a black solid with a yield of 30%; **Mes_2_CzBTM** is a black-purple solid with a yield of 23%.

**Ph_2_CzBTM**: **MALDI-TOF** (*m*/*z*): [M]^+^ calcd. for C_37_H_22_Cl_4_N˙, 622.05; found, 622.05. **Elem. Anal.** calcd. for C_37_H_22_Cl_4_N˙ (%): C, 71.40; H, 3.56; N, 2.25. Found (%): C, 71.22; H, 3.41; N, 2.08. **IR (KBr)** 2924 (m), 2853 (w), 2369 (w), 2341 (w), 1635 (s), 1446 (m), 1386 (m), 1051(m), 880 (w), 757 (w), 565 (w), 478 (w).

**Mes_2_CzBTM: MALDI-TOF** (*m*/*z*): [M]^+^ calcd. for C_43_H_34_Cl_4_N˙, 706.14; found, 706.14. **Elem. Anal.** calcd. for C_43_H_34_Cl_4_N˙ (%): C, 73.10; H, 4.85; N, 1.98. Found (%): C, 73.11; H, 4.59; N, 1.96. **IR (KBr)** 2922 (m), 2857 (w), 2346 (w), 1699 (m), 1633 (m), 1494 (m), 1447 (s), 1390 (m), 1338 (w), 1192 (w), 1084 (w), 855 (w), 802 (w), 750 (m), 720 (m), 673(w).

### 3.2. Synthesis of Ph_2_PyIDBTM and Mes_2_PyIDBTM

The specific synthetic route is detailed in Appendix A. **Ph_2_PyIDBTM** is a black-purple solid with a yield of 21%; **Mes_2_PyIDBTM** is a brown-red solid with a yield of 15%.

**Ph_2_PyIDBTM: MALDI-TOF** (*m*/*z*): [M]^+^ calcd. for C_36_H_21_Cl_4_N_2_˙, 623.04; found, 623.05. **Elem. Anal.** calcd. for C_36_H_21_Cl_4_N_2_˙ (%): C, 69.36; H, 3.40; N, 4.49. Found (%): C, 69.55; H, 3.43; N, 4.21. **IR (KBr)** 3053 (w), 2925 (s), 2854 (m), 2360 (w), 1618 (w), 1573 (w), 1515 (m), 1458(s), 1431 (s), 1388 (m), 1326 (w), 1274 (w), 1195 (w), 1080 (w), 968 (w), 869(w), 800(m), 757(m), 694(m), 601(w).

**Mes_2_PyIDBTM: MALDI-TOF** (*m*/*z*): [M]^+^ calcd. for C_42_H_33_Cl_4_N_2_˙, 707.14; found, 707.14. **Elem. Anal.** calcd. for C_42_H_33_Cl_4_N_2_˙ (%): C, 71.30; H, 4.70; N, 3.96. Found (%): C, 71.05; H, 4.94; N, 3.87. **IR (KBr)** 2921 (m), 2854 (w), 2362 (w), 1618 (m), 1571 (w), 1506 (m), 1458(s), 1429 (s), 1384 (m), 1326 (w), 1269 (w), 1216 (w), 1189 (w), 1033 (w), 852(w), 800(m), 742(m), 609(w).

## 4. Conclusions

By introducing phenyl and 2,4,6-trimethylphenyl groups into the structures of CzBTM and PyIDBTM, we successfully synthesized four new CzBTM-type radicals. The PL spectra of the four derivatives exhibited rare blueshifts in luminescent radicals, especially with the introduction of the 2,4,6-trimethylphenyl group. An analysis of the energy levels of the frontier molecular orbitals from theoretical calculations and CV measurements demonstrated the reason for blueshifts. The introduction of phenyl derivatives mainly caused the increase in the MO energy levels of the radical side, equivalent to the weakening of the electron-donating ability from the carbazole or β-carboline. Additionally, the PLQYs and thermostability were also improved, particularly for **Mes_2_CzBTM** with a highest PLQY of 21% and a highest Td of 311 °C under nitrogen. The introduction of phenyl and 2,4,6-trimethylphenyl groups provides a new strategy for adjusting the luminescent color of CzBTM-type radicals to shorter wavelengths and improving the thermostability.

## Figures and Tables

**Figure 1 molecules-29-02900-f001:**
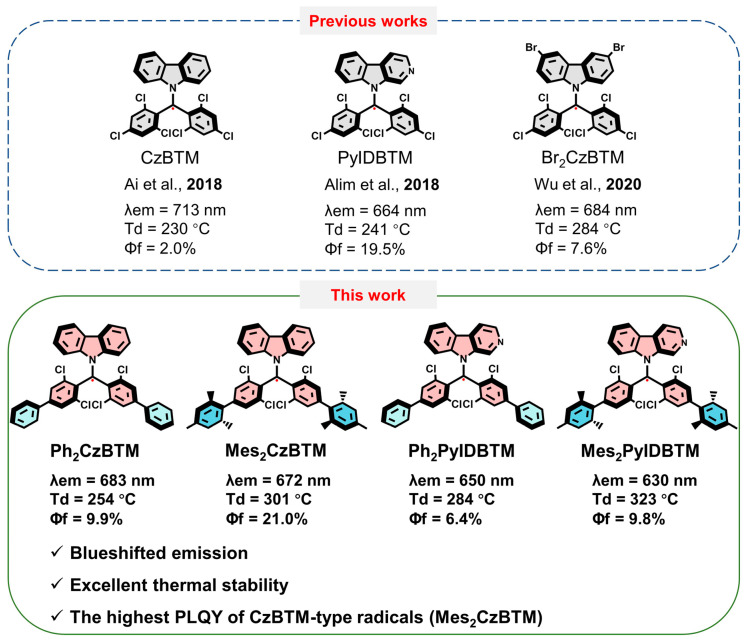
Molecular design strategy [31,32,33].

**Figure 2 molecules-29-02900-f002:**
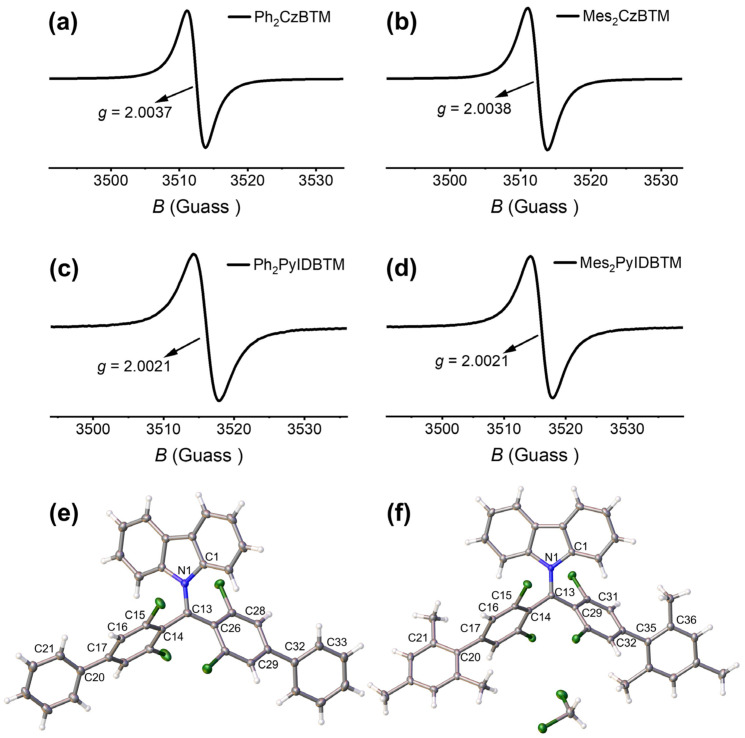
EPR spectrum of **Ph_2_CzBTM** (**a**), **Mes_2_CzBTM** (**b**), **Ph_2_PyIDBTM** (**c**), and **Mes_2_PyIDBTM** (**d**) in CH_2_Cl_2_ solvent (10^−3^ M) at room temperature; solved molecular structures of **Ph_2_CzBTM** (**e**) and **Mes_2_CzBTM** (**f**) from X-ray single-crystal diffraction.

**Figure 3 molecules-29-02900-f003:**
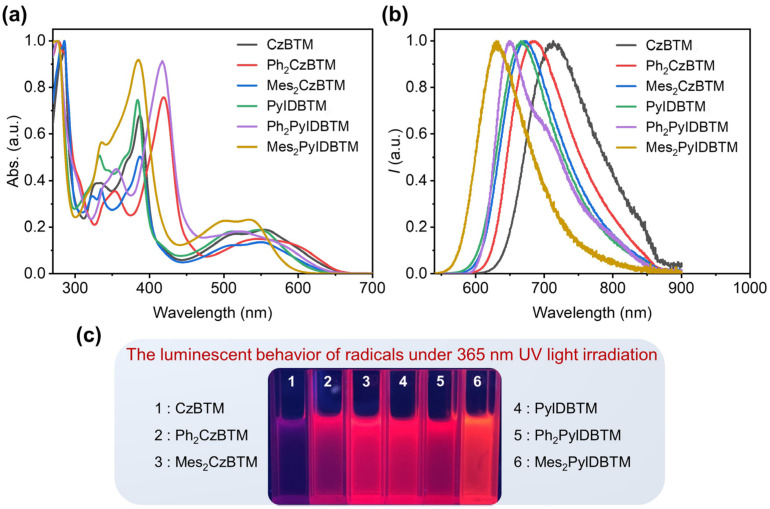
(**a**) The normalized UV-Vis absorption and PL spectra of CzBTM, PyIDBTM, **Ph_2_CzBTM**, **Mes_2_CzBTM**, **Ph_2_PyIDBTM,** and **Mes_2_PyIDBTM** in cyclohexane solution at room temperature; (**b**) the normalized PL spectra of CzBTM, PyIDBTM, **Ph_2_CzBTM**, **Mes_2_CzBTM**, **Ph_2_PyIDBTM,** and **Mes_2_PyIDBTM** in cyclohexane solution at room temperature; and (**c**) the luminescent behavior of radicals under 365 nm UV light irradiation in cyclohexane solvent.

**Figure 4 molecules-29-02900-f004:**
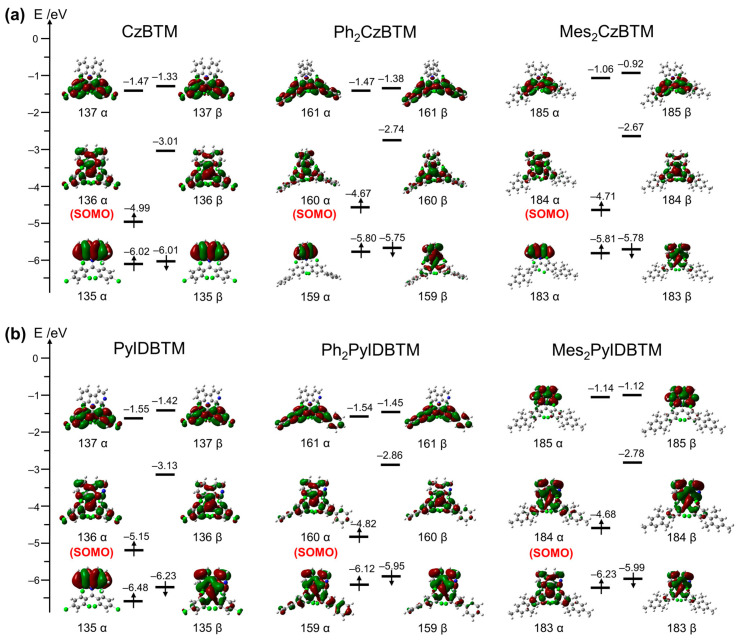
Frontier molecular orbitals calculated by DFT calculations. (**a**) CzBTM, **Ph_2_CzBTM**, and **Mes_2_CzBTM**; (**b**) PyIDBTM, **Ph_2_PyIDBTM**, and **Mes_2_PyIDBTM**. (Red and dark green distributions on molecule represent electron cloud distributions; green, blue, gray and white balls represent Cl, N, C and H atoms respectively.)

**Figure 5 molecules-29-02900-f005:**
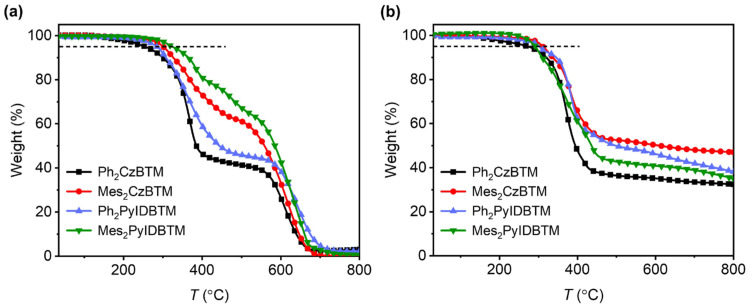
The TGA curves of four radicals with a heating rate of 10 °C/min under (**a**) air and (**b**) nitrogen. (The horizontal dashed black line represents the level of 5% mass loss.)

**Table 1 molecules-29-02900-t001:** Photophysical parameters of CzBTM-type radicals in cyclohexane.

Radicals	*λ_a_* (nm)	*λ_f_* (nm)	*Φ_f_* (%)	*τ* (ns)	*k*_r_ (s^−1^)	*k*_nr_ (s^−1^)
CzBTM	284, 387, 554 ^a^	713	2.0 ^a^	4.0 ^a^	0.5 × 10^7 a^	24.5 × 10^7 a^
**Ph_2_CzBTM**	277, 353, 545	683	9.9	9.3	1.1 × 10^7^	9.6 × 10^7^
**Mes_2_CzBTM**	284, 340, 543	672	21.0	15.2	1.4 × 10^7^	5.1 × 10^7^
PyIDBTM ^b^	260, 383, 550	664	19.5	12.8	1.4 × 10^7^	6.4 × 10^7^
**Ph_2_PyIDBTM**	276, 355, 522	650	6.4	6.4	1.0 × 10^7^	14.5 × 10^7^
**Mes_2_PyIDBTM**	276, 385, 535	630	9.8	8.6	1.1 × 10^7^	10.5 × 10^7^

^a^. Cited from ref. [31]; ^b^. Cited from ref. [33].

## Data Availability

The original contributions presented in the study are included in the article/Appendix A, further inquiries can be directed to the corresponding author.

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
