# Peer review of "Phenyl Derivatives Modulate the Luminescent Properties and Stability of CzBTM-Type Radicals"

_molecules, 2024, doi:10.3390/molecules29122900_

Round 1

Reviewer 1 Report

Comments and Suggestions for Authors

This paper reports synthesis and luminescent properties of four derivatives of CzPyBTM and PyIDBTM radicals. Substitution to phenyl or mesityl groups caused blue-shifts of emission and Mes2CzBTM showed the highest PLQE of 21%. In addition to optical measurements, DFT calculations and cyclic voltammetry were conducted and thermostabilities were found to be improved from the parent radicals. The properties of these radicals are very interesting and well described and explained in the manuscript. I recommend publication of this paper after some minor corrections.

1 (Line 17, Page 1). In general, the abbreviation PLQE stands for photoluminescence quantum efficiency and PLQY stands for photoluminescnece quantum yield. They are the same, but photoluminescence quantum yields (PLQE) is a bit confusing.

2 (Line 53 Page 2 and Line 174 Page 6)

The DFT results explanation seems to be more simply expressed that "the weakening of electron-accepting ability to radical" rather than "the weakening of electron-donating ability from carbazole or beta-carboline".

3 (Line 129, Page 5)

The lifetime of Mes2PyIDBTM is 9.8 ns in the text but 8.6 ns in the Table 1. Please correct one of them.

4 (Page 7, line 201 and supplementary infomation)

The reduction potential of CzBTM is -0.92V. Please correct the text.

Generally speaking, to use the onset potential for the redox potential of the compound is not appropriate. In the case of reversible cyclic votammogram, the bisection point of the reduction and oxidation peak potentials is the redox potential at [A]=[Ae-] stoichiometry.  That is one of the reason why you largely underestimate the energy-gap between alpha-SOMO and beta-SUMO. But it could not be helped since you could only compare the onset potentials to the previous literatures.

Reviewer 2 Report

Comments and Suggestions for Authors

Ai and coworkers report four blue-shifting luminescent radicals by substituting one of the chlorides on the benzene ring with a phenyl or a mesityl group from the previously reported luminescent radicals. This work, an extension of the previous work from these authors, has been done thoroughly using a combination of various techniques, from EPR to DFT calculations. I thought that the most interesting aspect of this work was the origin of blue-shifting, which appeared to differ between the two backbones (CzBTM and PylDBTM). As such, the substitution by the phenyl or the mesityl group would likely have different effects, such as whether the torsional strain plays a significant role. My main concern about this manuscript is the scope of the studies, and two in each class of luminescent radicals seemed too narrow to delineate the trend or the effect. I would recommend that authors perform DFT studies on the phenyl- or mesityl-substituted at more positions. Then, based on the calculation results, the authors could synthesize some representative radicals to verify the predictions. The manuscript is publishable in Molecules after the authors address the comments.

Page 3, Figure 2f: is CH2Cl2 part of the crystal?

Page 3, lines 84-88: the descriptions of the torsion and dihedral angles are confusing. I would recommend the authors label the corresponding four atoms to describe the dihedral angles.

Reviewer 3 Report

Comments and Suggestions for Authors

Manuscript “Phenyl Derivatives Modulate the Luminescent Properties and Stability of CzBTM-type Radicals” by Quanquan Gou, Jiahao Guan, Lintao Zhang, Xin Ai * presents the synthesis of four new stable organic radicals, as well as the results of studying their spectroscopic, thermal, photophysical and electrochemical properties. For the latter, the transition energies of D1 were determined using time-dependent DFT calculations. Theoretical calculations were carried out to define the frontier molecular orbitals all studied radicals. The best photoluminescence quantum yield of 21% was found for the 2,4,6-trimethylphenyl carbazolyl-containing radical (Mes2CzBTM). The paper is well composed and would be interested for the readers, but, unfortunately, there are some problems with the characterization of new radicals: a big disadvantage is the lack of the CHN analysis data, melting point values, and magnetochemical data for all four new radicals. Especially, PXRD are needed to prove the phase purity for the radicals with defined crystal structure, while for compounds without a defined XRD structure, magnetic measurement data is necessary to confirm the absence of diamagnetic impurities, due to the possible death of the radical center, while for compounds without a defined XRD structure, measurement data is required to confirm the absence of diamagnetic decomposition products of a radical. The manuscript cannot be published in its present form, as it requires revision.

The full list of comments is below.

In the introduction add the references for the next articles: http://dx.doi.org/10.1039/D0QM00992J, http://dx.doi.org/10.1039/C8TC04298E, http://dx.doi.org/10.1039/D0CY01339K

Do not use abbreviations in the text of the abstract. It is better to do it in the main text of the paper. The full IUPAC names of substances should be given in the abstract.

Lines 30-31: Usually, in most reported research, in a majority…

Generally, in most studies, the radical efficiency was improved by the introduction of electron-donating substituents.

31-36 The substituents that were introduced into the radicals were not only relatively simple donor groups such as indolyl [15], carbazolyl [26] and triphenylaminyl [27,28], but also much more complex substituted carbazole derivatives [29,30], including its dendritic derivatives [31]. It has been shown that this strategy is not only an effective way to improve the luminescent properties and radical stability, but also results in red-shifted emission spectra.

37 Give the full name of the radicals and put the abbreviations TTM, PyBTM and CzBTM in brackets.

37-38 Why is it necessary to achieve blue shift?

When hovering the mouse cursor over the all manuscript figures there is an insertion with Chinese hieroglyphics on each of them, please correct it.

9-(diphenylmethyl)-9H-carbazole

Figure 4 is too small, split into two figures and make them in full size. Figure 4. Frontier molecular orbitals calculated by DFT calculations. Make a separate image for each compound. Unify images a) and b) in Figure 5 using the same interval ranges for the Y axis.

43-45: Another successful approach to obtain blue-shifted emission spectra is the introduction of a nitrogen atom into the carbazolyl moiety of CzBTM, which leads to a weakening of the electron-donating property of carbazolyl.

46 …a different strategy was applied to change performance…

47 Given that CzBTM is itself a donor-acceptor molecule…

48 site or cite? What do you mean?

54 carbozolyl

147-148: were performed to understand the frontier molecular orbitals (MOs) of CzBTM-type radicals.

Line 273. Use the preposition "of" instead of  "about".

Decipher LDA in “Synthesis of compound 2”.

Combine the data from Table S1 and Table S2 into one table for easy comparison.

In Figure S1, replace the & sign with OR.

Provide a complete procedure for synthesizing compounds 7 and 9.

Since compounds 6-9 are radicals, the use of NMR spectroscopy is not able to verify the purity of the substances requires elemental analysis.

Correct in SM:

1.1 Sytheticprocdures……………………………………..……………………………  ………………………2

3. Absorption and fluorescence spectra of radicals in various solution …………………...…………………..13

Figure S4. FT-IR. Spectra are too small; enlarge each spectrum to make it the width of the page. Add wave number values below the peaks. Specify the conditions of the FT-IR experiment in experimental part. Indicate the values of the absorption bands in the FT-IR spectra of the newly obtained radicals.

There are too many curves in Figure 3!!! This makes it very difficult to see. Separate the UV-Vis absorption and PL spectra, and present each type of spectral information in a separate figure. Enlarge each spectrum to make it the width of the page.

Figures S9, S10. Separate the UV-Vis absorption and PL spectra, and present each type of spectral information in a separate figure, enlarge each spectrum to make it the width of the page.

In Figure S12, change the color of the positive spin density (green) to avoid confusion with the green chlorine atoms and make the molecules larger.

Correct Eeperimental in Table S6.

For the PyIDBTM radical, the redox potential values given in Table S6 need to be recalculated, since a different reference electrode was used.

Table S1, S2 put the name and abbreviation of the radicals

Try to fix or explain the level C alerts:

2341126

ABSTY02_ALERT_1_C  An _exptl_absorpt_correction_type has been given without a literature citation. This should be contained in the _exptl_absorpt_process_details field. Absorption correction given as multi-scan

PLAT905_ALERT_3_C Negative K value in the Analysis of Variance ...     -0.327 Report

PLAT911_ALERT_3_C Missing FCF Refl Between Thmin & STh/L=    0.600          4 Report

               -9  0 20,   4  2 24,   0  0 28,  -1  0 30,            

2341148

PLAT911_ALERT_3_C Missing FCF Refl Between Thmin & STh/L=    0.600          5 Report

 -3 -9  3,  -3 -9  4,   5-14  6,   0 -2  6,  -6  8  9,            Comments on the Quality of English Language

There are minor inaccuracies in grammar, syntax and style.
